# Seq2Slate: Re-ranking and Slate Optimization with RNNs

## Abstract

Ranking is a central task in machine learning and information retrieval. In this task, it is especially important to present the user with a slate of items that is appealing as a whole. This in turn requires taking into account interactions between items, since intuitively, placing an item on the slate affects the decision of which other items should be placed alongside it. In this work, we propose a sequence-to-sequence model for ranking called *seq2slate*. At each step, the model predicts the next item to place on the slate given the items already selected. The recurrent nature of the model allows complex dependencies between items to be captured directly in a flexible and scalable way. We show how to learn the model end-to-end from weak supervision in the form of easily obtained click-through data. We further demonstrate the usefulness of our approach in experiments on standard ranking benchmarks as well as in a real-world recommendation system.

## 1 Introduction

Ranking a set of candidate items is a central task in machine learning and information retrieval. Many existing ranking systems are based on pointwise estimators, where the model assigns a score to each item in a candidate set and the resulting *slate* is obtained by sorting the list according to item scores (Liu et al., 2009). Such models are usually trained from click-through data to optimize an appropriate loss function (Joachims, 2002). This simple approach is computationally attractive as it only requires a sort operation over the candidate set at test (or serving) time, and can therefore scale to large problems. On the other hand, in terms of modeling, pointwise rankers cannot easily express dependencies between ranked items. In particular, the score of an item (e.g., its probability of being clicked) often depends on the other items in the slate and their joint placement. Such interactions between items can be especially dominant in the common case where display area is limited or when strong position bias is present, so that only a few highly ranked items get the user's attention. In this case it may be preferable, for example, to present a *diverse* set of items at the top positions of the slate in order to cover a wider range of user interests.

A significant amount of work on learning-to-rank does consider interactions between ranked items when *training* the model. In *pairwise* approaches a classifier is trained to determine which item should be ranked first within a pair of items (e.g., Herbrich et al., 1999; Joachims, 2002; Burges et al., 2005). Similarly, in *listwise* approaches the loss depends on the full permutation of items (e.g., Cao et al., 2007; Yue et al., 2007). Although these losses consider inter-item dependencies, the ranking function itself is pointwise, so at inference time the model still assigns a score to each item which does not depend on scores of other items.

There has been some work on trying to capture interactions between items in the ranking scores themselves (e.g., Qin et al., 2008; 2009; Zhu et al., 2014; Rosenfeld et al., 2014; Dokania et al., 2014). Such approaches can, for example, encourage a pair of items to appear next to (or far from) each other in the resulting ranking. Approaches of this type often assume that the relationship between items takes a simple form (e.g., submodular) in order to obtain tractable inference and learning algorithms. Unfortunately, this comes at the expense of the model's expressive power.

In this paper, we present a general, scalable approach to ranking, which naturally accounts for high-order interactions. In particular, we apply a *sequence-to-sequence (seq2seq)* model (Sutskever et al., 2014) to the ranking task, where the input is the list of candidate items and the output is the resulting ordering. Since the output sequence corresponds to ranked items on the slate, we call this model *sequence-to-slate (seq2slate)*. The order in which the input is processed can significantly

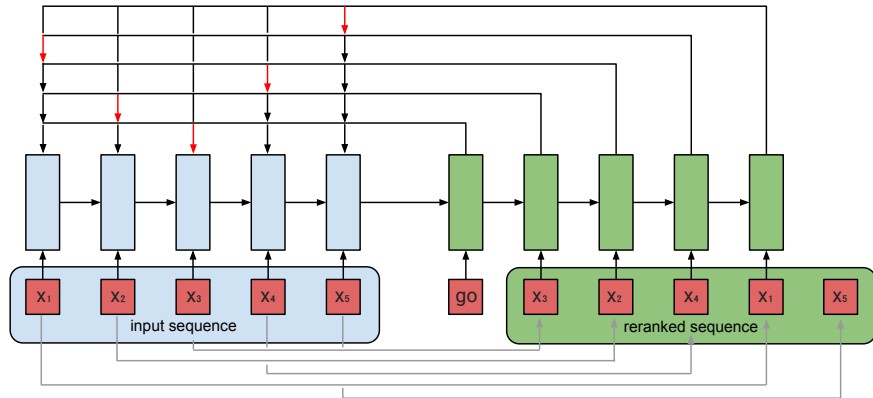

Figure 1: The seq2slate pointer network architecture for ranking.

affect the performance of such models (Vinyals et al., 2016). For this reason, we often assume the availability of a base (or "production") ranker with which the input sequence is ordered (e.g., a simple pointwise method that ignores the interactions we seek to model), and view the output of our model as a *re-ranking* of the items.

To address the seq2seq problem, we build on the recent success of *recurrent neural networks (RNNs)* in a wide range of applications (e.g., Sutskever et al., 2014). This allows us to use a deep model to capture rich dependencies between ranked items, while keeping the computational cost of inference manageable. More specifically, we use *pointer networks*, which are seq2seq models with an attention mechanism for pointing at positions in the input (Vinyals et al., 2015b). We show how to train the network end-to-end to directly optimize several commonly used ranking measures. To this end, we adapt RNN training to use weak supervision in the form of click-through data obtained from logs, instead of relying on ground-truth rankings, which are much more expensive to obtain. Finally, we demonstrate the usefulness of the proposed approach in a number of learning-to-rank benchmarks and in a large-scale, real-world recommendeation system.

## 2 RANKING AND SLATE OPTIMIZATION AS SEQUENCE PREDICTION

The *ranking problem* is that of computing a ranking of a set of items (or ordered list or *slate*) given some query or context. We formalize the problem as follows. Assume a set of $n$ items, each represented by a feature vector $x_i \in \mathbb{R}^m$ (which may depend on a query or context). Let $\pi \in \Pi$ denote a permutation of the items, where each $\pi_j \in \{1, \ldots, n\}$ denotes the index of the item in position $j$. Our goal is to predict the output ranking $\pi$ given the input items $x$. For instance, given a specific user query, we might want to return an ordered set of music recommendations from a set of candidates that maximizes some measure of user engagement (e.g., number of tracks played).

In the seq2seq framework, the probability of an output permutation, or slate, given the inputs is expressed as a product of conditional probabilities according to the chain rule:

$$p(\pi|x) = \prod_{j=1}^{n} p(\pi_j|\pi_1, \ldots, \pi_{j-1}, x) \,, \tag{1}$$

This expression is completely general and does not make any conditional independence assumptions. In our case, the conditional $p(\pi_j|\pi_{<j}, x) \in \Delta^n$ (a point in the $n$-dimensional simplex) models the probability of any item being placed at the $j$'th position in the ranking given the items already placed at previous positions. Therefore, this conditional exactly captures *all* high-order dependencies between items in the ranked list, including those due to diversity, similarity or other interactions.

Our setting is somewhat different than a standard seq2seq setting in that the output vocabulary is not fixed. In particular, the same index (position) is populated by different items in different instances (queries). Indeed, the vocabulary size $n$ itself may vary per instance in the common case where the number of items to rank can change. This is precisely the problem addressed by *pointer networks*, which we review next.

POINTER-NETWORK ARCHITECTURE FOR RANKING

We employ the *pointer-network architecture* of Vinyals et al. (2015b) to model the conditional $p(\pi_j|\pi_{<j}, x)$. A pointer network uses non-parametric softmax modules, akin to the attention mechanism of Bahdanau et al. (2015), and learns to point to items in its input sequence rather than predicting an index from a fixed-sized vocabulary.

Our *seq2slate* model, illustrated in Fig. 1, consists of two *recurrent neural networks* (RNNs): an encoder and a decoder, both of which use Long Short-term Memory (LSTM) cells (Hochreiter and Schmidhuber, 1997). At each encoding step $i \leq n$, the encoder RNN reads the input vector $x_i$ and outputs a $d$-dimensional vector $e_i$, thus transforming the input sequence $\{x_i\}_{i=1}^n$ into a sequence of latent memory states $\{e_i\}_{i=1}^n$. At each decoding step $j$, the decoder RNN outputs a $d$-dimensional vector $d_j$ which is used as a query in our attention function. The attention function takes as input the query $d_j \in \mathbb{R}^d$ and the set of latent memory states computed by the encoder $\{e_i\}_{i=1}^n$ and produces a probability distribution over the next item to include in the output sequence as follows:

$$s_i^j = v^\top \tanh\left(W_{enc} \cdot e_i + W_{dec} \cdot d_j\right) \tag{2}$$

$$p_\theta(\pi_j = i | \pi_{<j}, x) \equiv p_i^j = \begin{cases} e^{s_i^j} / \sum_{k \notin \pi_{<j}} e^{s_k^j} & \text{if } i \notin \pi_{<j} \\ 0 & \text{if } i \in \pi_{<j} \end{cases},$$

where $W_{enc}, W_{dec} \in \mathbb{R}^{d \times d}$ and $v \in \mathbb{R}^d$ are learned parameters in our network, denoted collectively by parameter vector $\theta$. The probability $p_i^j = p_\theta(\pi_j = i | \pi_{<j}, x)$, is obtained via a softmax over the remaining items and represents the degree to which the model points to input $i$ at decoding step $j$. To output a permutation, the $p_i^j$ are set to 0 for items $i$ that already appear in the slate. Once the next item $\pi_j$ is selected, typically greedily or by sampling (see below), its embedding $x_{\pi_j}$ is fed as input to the next decoder step. The input of the first decoder step is a learned $d$-dimensional vector, denoted as $go$ in Fig. 1. Importantly, $p_\theta(\pi|x)$ is differentiable for ant fixed permutation $\pi$ which allows gradient-based learning (see Section 3).

We note the following. (i) The model makes no explicit assumptions about the type of interactions between items. If the learned conditional in Eq. (2) is close to the true conditional in Eq. (1), then the model can capture rich interactions—including diversity, similarity or others. We demonstrate this flexibility in our experiments (Section 4). (ii) $x$ can represent either raw inputs or embeddings thereof, which can be learned together with the sequence model. (iii) The computational cost of inference, dominated by the sequential decoding procedure, is $O(n^2)$, which is standard in seq2seq models with attention. We also consider a computationally cheaper single-step decoder with linear cost $O(n)$, which outputs a single vector $p^1$, from which we obtain $\pi$ by sorting the values (similarly to pointwise ranking).

## 3 TRAINING WITH CLICK-THROUGH DATA

We now turn to the task of training the seq2slate model from data. A typical approach to learning in ranking systems is to run an existing ranker "in the wild" and log click-through data, which are then used to train an improved ranking model. This type of training data is relatively inexpensive to obtain, in contrast to human-curated labels such as relevance scores, ratings, or rankings (Joachims, 2002).

Formally, each training example consists of a sequence of items $\{x_1, \ldots, x_n\}$ and binary labels $(y_1, \ldots, y_n)$, with $y_i \in \{0, 1\}$, representing user feedback (e.g., click/no-click). Our approach easily extends to more informative feedback, such as the level of user engagement with the chosen item (e.g., time spent), but to simplify the presentation we focus on the binary case. Our goal is to learn the parameters $\theta$ of $p_\theta(\pi_j|\pi_{<j}, x)$ (Eq. (2)) such that permutations $\pi$ corresponding to "good" rankings are assigned high probabilities. Various performance measures $\mathcal{R}(\pi, y)$ can be used to evaluate the quality of a permutation $\pi$ given the labels $y$, for example, mean average precision (MAP), precision at $k$, or normalized discounted cumulative gain at $k$ (NDCG@k). Generally speaking, permutations where the positive labels rank higher are considered better.

In the standard seq2seq setting, models are trained to maximize the likelihood of a target sequence of tokens given the input, which can be done by maximizing the likelihood of each target token given the previous target tokens using Eq. (1). During training, the model is typically fed the ground-truth tokens as inputs to the next prediction step, an approach known as *teacher forcing* (Williams and Zipser, 1989). Unfortunately, this approach cannot be applied in our setting since we only have access to weak supervision in the form of labels $y$ (e.g clicks), rather than ground-truth permutations. Instead, we show how the seq2slate model can be trained directly from the labels $y$.

### 3.1 Training using REINFORCE

One potential approach, which has been applied successfully in related tasks (Bello et al., 2017; Zhong et al., 2017), is to use *reinforcement learning (RL)* to directly optimize for the ranking measure $\mathcal{R}(\pi, y)$. In this setup, the objective is to maximize the expected ranking metric obtained by sequences sampled from our model: $\mathbb{E}_{\pi \sim p_\theta(.|x)}[\mathcal{R}(\pi, y)]$. One can use policy gradients and stochastic gradient ascent to optimize $\theta$. The gradient is formulated using the popular REINFORCE update (Williams, 1992) and can be approximated via Monte-Carlo sampling as follows:

$$\nabla_\theta \mathbb{E}_{\pi \sim p_\theta(.|x)}[\mathcal{R}(\pi, y)] = \mathbb{E}_{\pi \sim p_\theta(.|x)}\Big[\mathcal{R}(\pi, y)\nabla_\theta \log p_\theta(\pi \mid x)\Big]$$

$$\approx \frac{1}{B}\sum_{k=1}^{B}\Big(\mathcal{R}(\pi_k, y_k) - b(x_k)\Big)\nabla_\theta \log p_\theta(\pi_k \mid x_k) , \tag{3}$$

where $k$ indexes ranking instances in a batch of size $B$, $\pi_k$ are permutations drawn from the model $p_\theta$ and $b(x)$ denotes a baseline function that estimates the expected rewards to reduce the variance of the gradients.

### 3.2 Supervised training

RL, however, is known to be a challenging optimization problem and can suffer from sample inefficiency and difficult credit assignment. As an alternative, we propose *supervised learning* using the labels $y$. In particular, rather than waiting until the end of the output sequence (as in RL), we wish to give feedback to the model at each decoder step.

Consider the first step, and recall that the model assigns a score $s_i$ to each item in the input. We define a per-step loss $\ell(s, y)$ which essentially acts as a multi-label classification loss with labels $y$ as ground truth. Two natural, simple choices for $\ell$ are cross-entropy loss and hinge loss:

$$\ell_{xent}(s, y) = -\sum_i \hat{y}_i \log p_i \tag{4}$$

$$\ell_{hinge}(s, y) = \max\{0, 1 - \min_{i:y_i=1} s_i + \max_{j:y_j=0} s_j\} ,$$

where $\hat{y}_i = y_i / \sum_j y_j$, and $p_i$ is a softmax of $s$, similar to Eq. (2). Intuitively, with cross-entropy loss we try to assign high probabilities to positive labels (see also Kurata et al., 2016), while hinge loss is minimized when scores of items with positive labels are higher than scores of those with negative labels. Notice that both losses are convex functions of the scores $s$. To improve convergence, we consider a smooth version of the hinge-loss where the maximum and minimum are replaced by their smooth counterparts: `smooth-max`$(s; \gamma) = \frac{1}{\gamma} \log \sum_i e^{\gamma s_i}$ (and smooth minimum is defined similarly, using $\min_i(s_i) = -\max_i(-s_i)$).

If we simply apply a per-step loss from Eq. (4) to all steps of the output sequence while reusing the labels $y$ at each step, then the loss is invariant to the actual output permutations (e.g., predicting a positive item at the beginning of the sequence has the same cost as predicting it at the end). Instead, we let the loss $\ell$ at each decoding step $j$ depend on the items already chosen, so no further loss is incurred after a label is predicted correctly. In particular, for a *fixed* permutation $\pi$, define the sequence loss:

$$\mathcal{L}_\pi(S, y) = \sum_{j=1}^{n} w_j \, \ell_{\pi_{<j}}(s^j, y) , \tag{5}$$

where $S = \{s^j\}_{j=1}^n$, and $\ell_{\pi_{<j}}(s^j, y)$ depends only on the indices in $s^j$ and $y$ which are not in the prefix permutation $\pi_{<j} = (\pi_1, \ldots, \pi_{j-1})$ (see Eq. (4)). Including a per-step weight $w_j$ can encourage better performance earlier in the sequence (e.g., $w_j = 1/\log(j + 1)$). Furthermore, if optimizing for a particular slate size $k$ is desired, one can restrict this loss to just the first $k$ output steps.

#### Decoding policies during training

Since teacher-forcing is not an option, we resort to feeding the model its own previous predictions, as in Bengio et al. (2015); Ranzato et al. (2016). In this case, the permutation $\pi$ is not fixed, but rather depends on the scores $S$. Specifically, we consider two policies for producing a permutation during training, *sampling and greedy decoding*, and introduce their corresponding losses.

**Greedy policy** The greedy policy consists of selecting the item that maximizes $p_\theta(\cdot|\pi_{<j}, x)$ at every time step $j$. The resulting permutation $\pi^*$ then satisfies $\pi_j^* = \mathrm{argmax}_i\, p_\theta(\pi_j = i|\pi_{<j}^*)$ and our loss becomes $\mathcal{L}_{\pi^*}$. The greedy policy loss is not continuous everywhere since a small change in the scores $s$ may result in a jump between permutations, and therefore $\mathcal{L}_\pi$. Specifically, the loss is non-differentiable when any $s^j$ has multiple maximizing arguments. Outside this measure-zero subspace, the loss is continuous (almost everywhere), and the gradient is well-defined.

**Sampling policy** The sampling policy consists of drawing each $\pi_j$ from $p_\theta(\cdot|\pi_{<j}, x)$. The corresponding loss $\mathbb{E}[\mathcal{L}] = \sum_\pi p_\theta(\pi)\mathcal{L}_\pi(\theta)$ is differentiable everywhere since both $p_\theta(\pi)$ and $\mathcal{L}_\pi(\theta)$ are differentiable for any permutation $\pi$ (See appendix for a direct derivation of $\mathbb{E}[\mathcal{L}]$ as a function of $S$). In this case, the gradient is formulated as:

$$
\begin{aligned}
\nabla_\theta \mathbb{E}[\mathcal{L}(\theta)] &= \nabla_\theta \sum_\pi p_\theta(\pi)\mathcal{L}_\pi(\theta) \\
&= \sum_\pi [(\nabla_\theta p_\theta(\pi))\mathcal{L}_\pi(\theta) + p_\theta(\pi)(\nabla_\theta \mathcal{L}_\pi(\theta))] \\
&= \mathbb{E}_{\pi \sim p_\theta} [\mathcal{L}_\pi(\theta) \cdot \nabla_\theta \log p_\theta(\pi) + \nabla_\theta \mathcal{L}_\pi(\theta)] \ ,
\end{aligned}
\tag{6}
$$

which can be approximated by:

$$
\frac{1}{B}\sum_{k=1}^{B}\left[\left(\mathcal{L}_{\pi_k}(\theta) - b(x_k)\right)\nabla_\theta \log p_\theta(\pi_k \mid x_k) + \nabla_\theta \mathcal{L}_{\pi_k}(\theta)\right] \ ,
\tag{7}
$$

where $b(x_k)$ is a baseline that approximates $\mathcal{L}_{\pi_k}(\theta)$. Applying stochastic gradient descent intuitively decreases both the loss of any sample (right term) but also the probability of drawing samples with high losses (left term). Notice that our gradient calculation differs from scheduled sampling (Bengio et al., 2015) which instead computes the loss of the sampled sequences (right term) but ignores the probability of sampling high loss sequences (left term). We found it helpful to include both terms, which may apply more generally to training of sequence-to-sequence models (Bengio et al., 2015; Goyal et al., 2016).

For both training policies, we minimize the loss via stochastic gradient descent over mini-batches in an *end-to-end* fashion.

## 4 EXPERIMENTAL RESULTS

We evaluate the performance of our seq2slate model on a collection of ranking tasks. In Section 4.1 we use learning-to-rank benchmark data to study the behavior of the model. We then apply our approach to a large-scale commercial recommendation system and report the results in Section 4.2.

**Implementation Details** We set hyperparameters of our model to values inspired by the literature. All experiments use mini-batches of 128 training examples and LSTM cells with 128 hidden units. We train our models with the Adam optimizer (Kingma and Ba, 2014) and an initial learning rate of 0.0003 decayed every 1000 steps by a factor of 0.96. Network parameters are initialized uniformly at random in $[-0.1, 0.1]$. To improve generalization, we regularize the model by using dropout with probability of dropping $p_{dropout} = 0.1$ and L2 regularization with a penalty coefficient $\lambda = 0.0003$. Unless specified otherwise, all results use supervised training with cross-entropy loss $\ell_{xent}$ and the sampling policy. At inference time, we report metrics for the greedy policy. We use an exponential moving average with a decay rate of 0.99 as the baseline $b(x)$ in Eq. (3) and Eq. (6). When training the seq2slate model with REINFORCE, we use $\mathcal{R} = \texttt{NDGC@10}$ as the reward function and do not regularize the model. We also considered a bidirectional encoder RNN (Schuster and Paliwal, 1997) but found that it did not lead to significant improvements in our experiments.

### 4.1 LEARNING-TO-RANK BENCHMARKS

To understand the behavior of the proposed model, we conduct experiments using two learning-to-rank datasets. We use two of the largest publicly available benchmarks: the Yahoo Learning to Rank Challenge data (set 1),[1] and the Web30k dataset.[2] All context (query) features are embedded within the item feature vectors themselves.

---

[1] https://webscope.sandbox.yahoo.com/catalog.php?datatype=c
[2] https://www.microsoft.com/en-us/research/project/mslr/

| Ranker | Yahoo | | | Web30k | | |
|---|---|---|---|---|---|---|
| | MAP | NDCG@5 | NDCG10 | MAP | NDCG@5 | NDCG@10 |
| seq2slate | **0.67** | **0.69** | **0.75** | **0.51** | **0.53** | **0.59** |
| AdaRank | 0.58 | 0.61 | 0.69 | 0.37 | 0.38 | 0.46 |
| Coordinate Ascent | 0.49 | 0.51 | 0.59 | 0.31 | 0.33 | 0.39 |
| LambdaMART | 0.58 | 0.61 | 0.69 | 0.42 | 0.46 | 0.52 |
| ListNet | 0.49 | 0.51 | 0.59 | 0.43 | 0.47 | 0.53 |
| MART | 0.58 | 0.60 | 0.68 | 0.39 | 0.42 | 0.48 |
| Random Forests | 0.54 | 0.57 | 0.65 | 0.36 | 0.39 | 0.45 |
| RankBoost | 0.50 | 0.52 | 0.60 | 0.24 | 0.25 | 0.30 |
| RankNet | 0.54 | 0.57 | 0.64 | 0.43 | 0.47 | 0.53 |

Table 1: Performance of seq2slate and other baselines on data generated with diverse-clicks.

We adapt the procedure proposed by Joachims et al. (2017) to generate click data. The original procedure is as follows: first, a base ranker is trained from the raw data. We select this base ranker by training all models in the RankLib package,[3] and selecting the one with the best performance on each data set (MART for Yahoo and LambdaMART for Web30k). We generate an item ranking using the base model, which is then used to generate training data by simulating a user "cascade" model: a user observes each item with decaying probability $1/i^\eta$, where $i$ is the base rank of the item and $\eta$ is a parameter of the generative model. This simulates a noisy sequential scan. An observed item is clicked if its ground-truth relevance score is above a threshold (relevant: $\{2, 3, 4\}$, irrelevant: $\{0, 1\}$), otherwise no click is generated.

To introduce high-order interactions, we augment the above procedure as follows, creating a generative process dubbed *diverse-clicks*. When observing a relevant item, the user will only click if it is not too similar to previously clicked items (i.e, diverse enough), thus reducing the total number of clicks. Similarity is defined as being in the smallest $q$ percentile (i.e., $q = 0.5$ is the median) of Euclidean distances between pairs of feature vectors within the same ranking instance: $d_{ij} = \|x_i - x_j\|$. We use $\eta = 0$ (no decay, since clicks are sparse anyway due to the diversity term) and $q = 0.5$. This modification to the generative model is essential for our purpose as the original data does not contain explicit inter-item dependencies. We also discuss variations of this model below.

Using the generated training data, we train both our seq2slate model and baseline rankers from the RankLib package: AdaRank (Xu and Li, 2007), Coordinate Ascent (Metzler and Croft, 2007), LambdaMART (Wu et al., 2010), ListNet (Cao et al., 2007), MART (Friedman, 2001), Random Forests (Breiman, 2001), RankBoost (Freund et al., 2003), RankNet (Burges et al., 2005). Some of these baselines use deep neural networks (e.g., RankNet, ListNet), so they are strong state-of-the-art models with comparable complexity to seq2slate. The results in Table 1 show that seq2slate significantly outperforms all the baselines, suggesting that it can better capture and exploit the dependencies between items in the data.

To better understand the behavior of the model, we visualize the probabilities of the attention from Eq. (2) for one of the test instances in Fig. 2. Interestingly, the model produces slates that are close to the input ranking, but with some items demoted to lower positions, presumably due to the interactions with previous items.

We next consider several variations of the generative model and of the seq2slate model itself. Results are reported in Table 2. The rank-gain metric per example is computed by summing the positions change of all positive labels in the re-ranking, and this is averaged over all examples (queries).

**Comparison of training variants** In Table 2, we compare the different training variants outlined in Section 3, namely cross entropy with the greedy or sampling policy, a smooth hinge loss with $\gamma = 1.0$, and REINFORCE. We find that supervised learning with cross entropy generally performs best, with the smooth hinge loss doing slightly worse. Our weakly supervised training methods have positive rank gain on all datasets, meaning they improve over the base ranker. Results from Table 2 (see also Table 5 in the appendix) suggest that training with REINFORCE yields comparable results on Yahoo but significantly worse results on the more challenging Web30k dataset. We find no significant difference in performance between relying on the greedy and sampling policies during training.

---

[3] https://sourceforge.net/p/lemur/wiki/RankLib/

| Ranker | Yahoo | | | | Web30k | | | |
|---|---|---|---|---|---|---|---|---|
| | MAP | NDCG@5 | NDCG@10 | rank-gain | MAP | NDCG@5 | NDCG@10 | rank-gain |
| seq2slate | **0.67** | **0.69** | **0.75** | **7.4** | **0.51** | **0.53** | **0.59** | **18.3** |
| Greedy policy | 0.66 | 0.69 | 0.75 | 7.2 | 0.50 | 0.52 | 0.59 | 18.3 |
| smooth-hinge | 0.66 | 0.69 | 0.75 | 7.1 | 0.49 | 0.51 | 0.58 | 17.9 |
| RL | 0.66 | 0.68 | 0.75 | 5.7 | 0.44 | 0.47 | 0.53 | -0.5 |
| one-step decoder | 0.66 | 0.69 | 0.75 | 6.4 | 0.49 | 0.51 | 0.58 | 16.5 |
| shuffled data | 0.61 | 0.64 | 0.71 | – | 0.36 | 0.36 | 0.44 | – |
| base ranker (no-op) | 0.58 | 0.61 | 0.69 | 0 | 0.45 | 0.48 | 0.54 | 0 |

Table 2: Comparison of model and data variants for seq2slate on data generated with diverse-clicks.

| Ranker | MAP | NDCG@5 | NDCG@10 | rank-gain |
|---|---|---|---|---|
| one-step decoder | +26.79% | +10.69% | +40.67% | 0.83 |
| seq2slate | +31.32% | +14.47% | +45.77% | 1.087 |

Table 3: Performance compared to a competitive base production ranker on real data.

**One-step decoding**   We compare seq2slate to the model which uses a single decoding step, referred to as *one-step decoder* (see Section 2). In Table 2 we see that this model has comparable performance to the sequential decoder. This suggests that when inference time is crucial, as in many real-world systems, one might prefer the faster single-shot option. One possible explanation for the comparable performance of the one-step decoder is that the interactions in our generated data are rather simple and can be effectively learned by the encoder. By contrast, in Section 4.2 we show that on more complex real-world data, sequential decoding can perform significantly better.

**Sensitivity to input order**   Previous work suggests that the performance of seq2seq models are often sensitive to the order in which the input is processed (Vinyals et al., 2016; Nam et al., 2017). To test this we consider the use of seq2slate without relying on the base ranker to order the input, but instead items are fed to the model in random order. The results in Table 2 (see *shuffled data*) show that the performance is indeed significantly worse in this case, which is consistent with previous studies. It suggests that reranking is an easier task than ranking from scratch.

**Adaptivity to the type of interaction**   To demonstrate the flexibility of seq2slate, we generate data using a variant of the diverse-clicks model above. In the *similar-clicks* model, the user also clicks on observed irrelevant items if they are similar to previously clicked items (increasing the number of total clicks). As above, we use the pairwise distances in feature space $d_{ij}$ to determine similarity. For this model we use $q = 0.5$, and $\eta = 0.3$ for Web30k, $\eta = 0.1$ for Yahoo, to keep the proportion of positive labels similar. The results in the appendix (see Table 4) show that seq2slate has comparable performance to the baseline rankers, with slightly better performance on the harder Web30k data. This demonstrates that our model can adapt to various types of interactions in the data.

## 4.2   REAL-WORLD DATA

We also apply seq2slate to a ranking problem from a large-scale commercial recommendation system. We train the model using massive click-through logs (comprising roughly $O(10^7)$ instances) with cross-entropy loss, the greedy policy, L2-regularization and dropout. The data has item sets of varying size, with an average $n$ of 10.24 items per example. We learn embeddings of the raw inputs as part of training. Table 3 shows the performance of seq2slate and the one-step decoder compared to the production base ranker on test data (of roughly the same size as the training data). Significant gains are observed in all performance metrics, with sequential decoding outperforming the one-step decoder. This suggests that sequential decoding may more faithfully capture complex dependencies between the items.

Finally, we let the learned seq2slate model run in a live experiment (A/B testing). We compute the click-through rate (CTR) in each position (#clicks/#examples) for seq2slate. The production base ranker serves traffic outside the experiment, and we compute CTR per position for this traffic as well. Fig. 3 shows the difference in CTR per position, indicating that seq2slate has significantly higher CTR in the top positions. This suggests that seq2slate indeed places items that are likely to be chosen higher in the ranking.

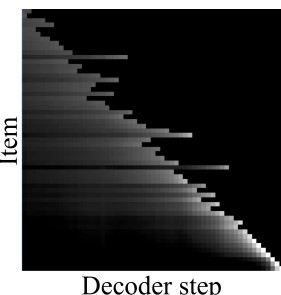

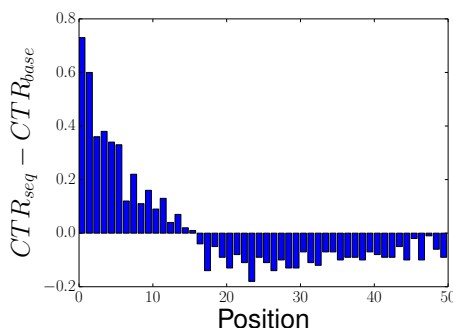

Figure 2: Visualization of attention probabilities on benchmark data. Intensities correspond to $p_i^j$ for each item $i$ in step $j$.

Figure 3: Difference in CTR per position between a seq2slate model and a base production ranker in a live experiment.

## 5 RELATED WORK

In this section we discuss additional related work. Our work builds on the recent impressive success of seq2seq models in complex prediction tasks, including machine translation (Sutskever et al., 2014; Bahdanau et al., 2015), parsing (Vinyals et al., 2015a), combinatorial optimization (Vinyals et al., 2015b; Bello et al., 2017), multi-label classification (Wang et al., 2016; Nam et al., 2017), and others. Our work differs in that we explicitly target the ranking task, which requires a novel approach to training seq2seq models from weak feedback (click-through data).

Most of the work on ranking mentioned above uses shallow representations. However, in recent years deep models have been used for information retrieval, focusing on embedding queries, documents and query-document pairs (Huang et al., 2013; Guo et al., 2016; Palangi et al., 2016; Wang and Klabjan, 2017; Pang et al., 2017) (see also recent survey by Mitra and Craswell (2017)). Rather than embedding individual items, in seq2slate a representation of the entire slate of items is learned and encoded in the RNN state. Moreover, learning the embeddings ($x$) can be easily incorporated into the training of the sequence model to optimize both simultaneously end-to-end.

Closest to ours is the recent work of Ai et al. (2018), where an RNN is used to encode a set of items for re-ranking. Their approach uses a single decoding step with attention, similar to our one-step decoder. In contrast, we use sequential decoding, which we find crucial in certain applications (see Section 4.2). Another important difference is that their training formulation assumes availability of full rankings or relevance scores, while we focus on learning from cheap click-through data.

Finally, Santa Cruz et al. (2017) recently proposed an elegant framework for learning permutations based on the so called Sinkhorn operator. Their approach uses a continuous relaxation of permutation matrices (i.e., the set of doubly-stochastic matrices). Later, Mena et al. (2018) combined this with a Gumbel softmax distribution to enable efficient learning. However, this approach is focused on reconstruction of scrambled objects, and it is not obvious how to extend it to our ranking setting, where no ground-truth permutation is available.

## 6 CONCLUSION

We presented a novel seq2slate approach to ranking sets of items. We found the formalism of pointer-networks particularly suitable for this setting. We addressed the challenge of training the model from weak user feedback to improve the ranking quality. Our experiments show that the proposed approach is highly scalable and can deliver significant improvements in ranking results.

Our work can be extended in several directions. In terms of architecture, we aim to explore the *Transformer* network (Vaswani et al., 2017) in place of the RNN. Several variants can potentially improve the performance of our model, including beam-search inference (Wiseman and Rush, 2016), and training with Actor-Critic (Bahdanau et al., 2017) or SeaRNN (Leblond et al., 2018) and it will be interesting to study their performance in the ranking setting. Finally, an interesting future work direction will be to study off-policy correction (Joachims et al., 2018) for seq2slate.

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

# A   DERIVATION OF THE EXPECTED LOSS

$$
\begin{aligned}
\mathbb{E}[\mathcal{L}] &= \sum_{\pi} p(\pi) \mathcal{L}_{\pi} \\
&= \sum_{\pi} p(\pi) \sum_{j} \ell_{\pi_{<j}} \\
&= \sum_{j} \sum_{\pi} p(\pi_{<j}) p(\pi_{\geq j} | \pi_{<j}) \ell_{\pi_{<j}} \\
&= \sum_{j} \sum_{\pi_{<j}} p(\pi_{<j}) \ell_{\pi_{<j}} \overbrace{\sum_{\pi_{\geq j}} p(\pi_{\geq j} | \pi_{<j})}^{1} \\
&= \sum_{j} \sum_{\pi_{<j}} \left( \prod_{k=1}^{j-1} e^{s_{\pi_k}^k} / \sum_{i \notin \pi_{<k}} e^{s_i^k} \right) \ell_{\pi_{<j}}(s^j, y) \quad .
\end{aligned}
$$

Since the terms are continuous (and smooth) in $S$ for all $j$ and $\pi_{<j}$, so is the entire function.

# B   ADDITIONAL EXPERIMENTAL RESULTS

| Ranker | Yahoo | | | Web30k | | |
|---|---|---|---|---|---|---|
| | MAP | NDCG@5 | NDCG@10 | MAP | NDCG@5 | NDCG@10 |
| seq2slate | 0.82 | 0.82 | 0.84 | **0.44** | **0.54** | **0.50** |
| AdaRank | 0.83 | 0.81 | 0.84 | 0.41 | 0.52 | 0.48 |
| Coordinate Ascent | 0.83 | 0.82 | 0.85 | 0.39 | 0.47 | 0.44 |
| LambdaMART | **0.84** | **0.83** | **0.85** | 0.41 | 0.52 | 0.48 |
| ListNet | 0.83 | 0.83 | 0.85 | 0.41 | 0.53 | 0.49 |
| MART | 0.83 | 0.82 | 0.85 | 0.41 | 0.52 | 0.48 |
| Random Forests | 0.83 | 0.82 | 0.84 | 0.40 | 0.48 | 0.45 |
| RankBoost | 0.83 | 0.83 | 0.85 | 0.38 | 0.43 | 0.41 |
| RankNet | 0.83 | 0.82 | 0.84 | 0.35 | 0.36 | 0.35 |

Table 4: Performance of seq2slate and other baselines on data generated with similar-clicks.

| Ranker | Yahoo | | | | Web30k | | | |
|---|---|---|---|---|---|---|---|---|
| | MAP | NDCG@5 | NDCG@10 | rank-gain | MAP | NDCG@5 | NDCG10 | rank-gain |
| seq2slate | **0.82** | **0.82** | **0.84** | **8.5** | **0.44** | **0.54** | **0.50** | **16.0** |
| Greedy policy | **0.82** | **0.82** | **0.84** | **8.5** | 0.44 | 0.54 | 0.50 | 15.9 |
| smooth-hinge | 0.80 | 0.80 | 0.82 | 7.7 | 0.44 | 0.54 | 0.50 | 15.9 |
| RL | **0.82** | **0.82** | **0.84** | **8.5** | 0.42 | 0.53 | 0.49 | -14.8 |
| one-step decoder | 0.81 | 0.81 | 0.82 | 7.7 | 0.44 | 0.53 | 0.49 | 15.5 |
| shuffled data | 0.80 | 0.80 | 0.81 | – | 0.40 | 0.44 | 0.42 | – |
| base ranker (no-op) | 0.78 | 0.76 | 0.79 | 0 | 0.43 | 0.53 | 0.49 | 0 |

Table 5: Comparison of model and data variants for seq2slate on data generated with similar-clicks.

