# OpenReview forum: "Seq2Slate: Re-ranking and Slate Optimization with RNNs"
_ICLR.cc/2019/Conference_

### Official Review · AnonReviewer1 · 2018-11-04
**Interesting application of pointer networks for re-ranking**

**Rating:** 7
**Confidence:** 5

**Review:**

If the stated revisions are incorporated into the paper, it will be a substantially stronger version. I'm leaning towards accepting the revised version -- all my concerns are addressed by the authors' comments.
---
The paper uses a Seq2Seq network to re-rank candidate items in an information retrieval task so as to account for inter-item dependencies in a weakly supervised manner. The gain from using such a re-ranker is demonstrated using synthetic experiments as well as a real-world experiment on live traffic to a recommender system.

Paragraph below eqn2: for *any* fixed permutation. Figure1 and notation indicates that, at each step of decoding, the selected input x_j is fed to the decoder. The text suggests the embedding of the input e_j is fed to the decoder (which is consistent with "go" being a d-dimensional vector, rather than the dimensionality of x).

Single step decoder with linear cost: Is there a missing footnote? If not, why call it p^1? Simpler notation to just call it p.
Eqn7: Role of baseline. In REINFORCE, b(x) is typically action-independent (e.g. approximating the state value function induced by the current policy). L_pi(theta) is action dependent (depends on the permutation sampled from P_theta). So, I'm unclear about the correctness of Eqn7 (does it yield an unbiased policy gradient?)

Eqn5: Expected some discussion about the mismatch between training loss (per-step cross entropy) vs. testing loss (e.g. NDCG@k). Does a suitable choice of w_j allow us to recover standard listwise metrics (that capture interactions, e.g. ERR)?

Expt implementation: Why was REINFORCE optimizing NDCG@10 not regularized?
Expt cascade click model: Expected an even simpler experiment to begin with; [Is the Seq2Slate model expressive enough to capture listwise metrics?] Since the relevances are available, we can check if Seq2Slate trained to the relevance labels yields NDCG performance comparable to LambdaMART, and whether it can optimize metrics like ERR.

Table1: On the test set, is NDCG&MAP being computed w.r.t the ground truth relevances? So, is the claim that Seq2Slate is more robust when clicks are noisy in a one-sided way (i.e. relevant items may not receive clicks)? Not clear how much of this benefit comes from a more expressive model to predict relevances (see suggested expt above) vs. Seq2Slate from clicks. NDCG & MAP definitely don't account for inter-item dependencies, so unclear what is being tested in this experiment.

For diverse-clicks, eta=0 while for similar-clicks, eta>0 (in a dataset dependent way). Why? Can expt numbers for the 3 choices of eta be added to the appendix? [Seems like cherry-picking otherwise]

Can Ai et al, 2018 be benchmarked on the current expt setup? Is it identical to the single-step decoder proposed in the paper?

Comment: Would be more accurate to call seq2slate a re-ranker throughout the text (in the abstract and intro, the claim is seq2slate is a ranker).

Expected to see training time and inference time numbers. Since Seq2Slate does extra computation on top of, e.g. LambdaMART, it is useful to know how scalable it can be during training, and when the extra perf is worth the O(n^2) or O(n) [for single-step decoding] during inference.

General comments:
Clarity: The paper is well written and easy to follow. There are a few notation choices that can be improved/clarified.
Originality: This work seems closely related to Ai et al, SIGIR 2018. Going from a single-shot decoding to sequential decoding is an incremental step; the real-world experiment seemed the most novel and compelling contribution (however, it is unclear how one can reproduce it).
Significance: The paper addresses a significant real-world problem. Many high-impact applications of ranking rely on being able to model inter-dependencies well.
Quality: The paper has interesting contributions, but can be substantially stronger (see some of the specific comments above). For instance, A careful study of the computational vs. performance trade-off, fine-grained comparison of different decoding architectures, better understanding of which architectural choices allow us to model any arbitrary ranking metric more effectively vs. which ones are more robust to click noise vs. which ones capture inter-item dependencies.

---

> ### Author Response · Authors · 2018-11-14
> **Clarifications**
>
> Thanks for the valuable feedback. We address specific points here and more general points in the common response.
>
> ## is x_j or e_j fed to the decoder?
> ===
> x_j is fed to the decoder. There is a typo in the draft: “go” is actually an m-dimensional vector. Thanks for pointing this out, we will fix this.
>
> ## why call it p^1? Simpler notation to just call it p. Is there a missing footnote?
> ===
> We use the notation defined in Eq 2 to refer to the vector of probabilities p^1_i for all items i as p^1. We denote by p the nXn matrix of probabilities for all positions and all items. (There is no missing footnote, the superscript is intended.) We’ll verify that the notation is explained clearly and make adjustments as needed.
>
> ## In REINFORCE, b(x) is typically action-independent… L_pi(theta) is action dependent… So, I'm unclear about the correctness of Eqn7 (does it yield an unbiased policy gradient?)
> ===
> Notice that in Eq 7 while L depends on \theta, the baseline b depends only on x_k. Here L takes the role of the reward in REINFORCE which is action-dependent. So this is indeed an unbiased variance reduction. See also reply to Reviewer 3.
>
> ## Expected some discussion about the mismatch between training loss vs. testing loss…
> ===
> Indeed, the sequence loss in Eq 5 is not exactly the same as the ranking measures used in evaluation. However, notice that any permutation that places the positive labels at the first positions gets 0 loss in all these cases, so in that sense the losses are aligned. There is no choice of weights w_j that achieves full correspondence. This is quite common for surrogate losses in machine learning. We will elaborate and clarify in the revision.
>
> ## Why was REINFORCE optimizing NDCG@10 not regularized?
> ===
> We observed that reinforce training was not prone to overfitting, which is generally expected since the policy gradients are noisy.
>
> ## Expected an even simpler experiment to begin with; [Is the Seq2Slate model expressive enough to capture listwise metrics?]
> ===
> The original benchmark data only include per-item relevance scores, without high-order interactions (unfortunately, publicly available data sets with high-order interactions do not seem to exist). In this case, the joint probability in Eq 1 is just p(\pi|x) = \prod_j p(\pi_j|x), and a pointwise ranker is optimal, so there would be no need for seq2slate. We take learning from click-through logs and modeling high-order interactions to be central to our approach, which in some sense makes this particular experiment less relevant/interesting as a means for evaluating seq2slate.
>
> ## Table1: On the test set, is NDCG&MAP being computed w.r.t the ground truth relevances? ... Not clear how much of this benefit comes from a more expressive model ... NDCG & MAP definitely don't account for inter-item dependencies, so unclear what is being tested in this experiment.
> ===
> All reported test measures are w.r.t. generated click data and not relevance scores. In terms of model expressivity, we note that some of the baselines (pointwise rankers) use deep neural nets with comparable complexity to seq2slate (in terms of number of layers and parameters), so we attribute the gains to better modeling.
> We disagree with the claim that “NDCG & MAP definitely don't account for inter-item dependencies”. As a counterexample, consider our “diverse-clicks” setting, where clicks happen on items that are different from each other. Placing diverse items at the top positions will result in better NDCG and MAP compared to placing items that are similar to each other in those positions. We will make a note of this in the revision.
>
> ## For diverse-clicks, eta=0 while for similar-clicks, eta>0 (in a dataset dependent way). Why? ... [Seems like cherry-picking otherwise]
> ===
> In diverse-clicks we noticed that if we used \eta>0 there were too many examples with all-zero labels (no clicks), which hindered training (since we cannot learn a good ranking from these examples). We then chose different values of \eta for each dataset such that the percentage of examples with no positive labels (clicks) remained small enough and roughly the same in all datasets:
>                            Yahoo                Web30K
> Diverse-clicks  1.15%(eta=0)    1.10%(eta=0)
> Similar-clicks  1.23%(eta=0.1)  1.14%(eta=0.3)
> That was the only criterion used to choose the value of \eta. We will clarify this in the revision.

---

> > ### Author Response · Authors · 2018-11-14
> > **Clarifications (cont)**
> >
> > ## Expected to see training time and inference time numbers.
> > ===
> > Unfortunately, it is hard to directly compare computation for pointwise models like LambdaMART to ours since we use the RankLib Java code to run the baselines while we use TensorFlow to train and test seq2slate. Also, while RankLib was run on a single machine, we used TensorFlow’s ability to easily parallelize training on multiple machines. We can, however, compare to the one-step decoder.
> >
> > On the benchmark datasets we observed a 4x decrease in training time and a 3x decrease in inference time for the one-step decoder compared to sequential decoding. For the real-world data, one-step decoding was 2.5x faster per iteration. We note that even with sequential decoding the runtime was not a bottleneck and we were able to train a seq2slate model on billions of examples in a couple of hours, and serve live traffic in milliseconds. For this reason we also did not make an effort to optimize the code, so the numbers above can probably be reduced significantly.
> > An important advantage of our approach is that computation is fixed in advance and the model is trained end-to-end. This allows us to avoid costly inference procedures used in previous work (e.g., Zhu et al., 2014; Rosenfeld et al., 2014; Dokania et al., 2014).
> >
> > We will add some discussion of computation to the revised paper so the reader develops a sense of the practicality of the approach.
> >
> > ## comparison of different architectures
> > ===
> > We compared a few architectures in our experiments, including bidirectional LSTM, stacked LSTM, and also played with the number of units in each layer of the deep network. We did see a degradation in performance when using 32 units or fewer, but otherwise the performance was not sensitive to the architecture choice and we only observed insignificant differences. We mentioned the results on bidirectional LSTM in the paper, but will add a discussion of the other results as well.

---

### Official Review · AnonReviewer2 · 2018-11-05
**Applies the pointer-network architecture to the re-ranking problem with promising empirical results on learning-to-rank testbeds and a real-world recommendation engine**

**Rating:** 6
**Confidence:** 4

**Review:**

The authors consider the problem of re-ranking an initial ranker that doesn’t consider interactions between items (e.g., a point-wise ranker) with a pointer-network approach that considers these interactions when re-ordering the input ranking. Notably, this is performed during decoding as opposed to {pairwise, list-wise} learning to rank approaches that consider interactions during training, but emit an item-wise score during inference. Operationally in practice, this has to be trained from click-through data for which the authors consider both a RL approach (Reinforce) and supervised training (a sequence-level hinge loss function) and decoded either with a single-step greedy policy or a sampling procedure. Experiments are conducted on learning-to-rank benchmarks where interactions are introduced to test the validity of the method and on a real-world, large-scale recommendation engine — showing solid improvements in both cases.

From a high-level perspective, the methodological innovation (a pointer-network trained on sequence loss from logged data), setting (re-ranking a slate to consider interactions), and empirical analyses are largely ‘incremental’ — although I think non-trivial to put together and the paper itself is well-written and fairly convincing. In framing the paper this way, I would have expected some comparison to sub-modular methods on the ‘diverse-clicks’ generated data for completeness, although I would be surprised if the Seq2Slate method doesn’t perform better (but all the more reason to conduct). In addition to reporting how to resolve some of the details of applying this, the most interesting results may very well be the real-world experiments as the result improvements are fairly impressive (such that I intend to play with this myself). Thus, as the focus is on details and empirical results over methodological innovation, this paper reads a bit like an industry-track paper — but I find the results interesting overall and am marginally inclined to accept.

Evaluating the paper along the requested dimensions:

= Quality: The paper clearly states its motivation, proposes a model, discusses practical issues, and provides convincing experiments (given the constraints of proprietary data, etc.). I didn’t observe any technical flaws and everything was relatively self-contained and easy to read. I could think of a few more experiments regarding submodular-based models, possibly different settings of the ‘diverse-click’ data for a sensitivity analysis, and a more direct comparison to [Ai, et al., SIGIR18], but this isn’t required to make the results sufficiently convincing. (6/10)

= Clarity: The paper is very clearly written. (7/10)

= Originality: This is the weakest aspect of the paper from a methodological perspective. It is a fairly straightforward application of pointer-networks. Even the path forward is fairly straightforward as outlined in the conclusion. One additional pointer that is methodologically similar, but for ‘discrete choice’ as opposed to re-ranking is [Mottini & Acuna-Agost, Deep Choice Model Using Pointer Networks for Airline Itinerary Prediction; KDD17] (which honestly, is probably a better venue for this specific work). Non-trivial and complete, but not particularly innovative. (5/10)

= Significance: Methodologically, probably will influence some work regarding re-ranking methodologically. From a practical perspective, seems very promising. A few more experiments would make this case stronger, but has real-world data. (6/10)

=== Pros ===
+ extends a widely used model (pointer-networks) to the re-ranking setting
+ discusses practical issues in getting this to work at scale
+ shows that it works in a real-world setting
+ contextualization within existing research shows good understanding of related work

=== Cons ===
- is a fairly direct application of pointer-networks with the innovation being in the details (i.e., is more of an ‘industry’ paper)
- additional experiments around ‘diverse-clicks’ settings (to see how smooth the performance curve) and submodular comparisons may have been interesting

In summary, I think there is room for improvement (some outlined in the conclusion), but is an interesting finding with promise that I plan to try myself. Thus, I lean toward an accept.

---

> ### Author Response · Authors · 2018-11-14
> **Clarifications**
>
> Thanks for the valuable feedback. We address specific points here and more general points in the common response.
>
> We certainly believe that the fact that our work has practical applications and demonstrated in a large-scale, real-world system adds to our scientific contribution and should be considered an advantage of our work. That said, we want to emphasize that the scientific contributions discussed in the common response, are critical to its success, broadly applicable and, as mentioned above, well-aligned with ICLR’s focus on learning representations. We don’t believe that the paper should be viewed as one intended for an “industry track.”
>
> Regarding experiments:
> * Comparison to sub-modular baseline
> We agree that this is an interesting baseline. We are not aware of publicly available code (we would appreciate any pointers in case we missed something), but will work to add such baseline in the revision.
> We emphasize that seq2slate is flexible and data-driven rather than modeling specific types of interactions, which is a key advantage of our approach. This allows us to avoid strong assumptions regarding the type of interactions between items made by a large number of previous approaches, and lets the model adapt to the type of interactions present in the data. For example, if one used a specific interaction model for ‘diverse-clicks’, then a different model would be required for the ‘similar-clicks’ data, a distinction not needed with seq2slate.
> * Different settings of the ‘diverse-click’ data for sensitivity analysis
> We compare ‘diverse-clicks’ to ‘similar-clicks’ in the experiments, which tests a different type of interaction. In our generative model we also had to make sure that the total number of clicks was suitable for training on the benchmark data, which restricted the range of values (see also reply to Reviewer 1 on the choice of \eta).
> * A more direct comparison to Ai, et al., 2018 -- see common reply.
>
> Thanks for the reference to Mottini & Acuna-Agost, we will include it.

---

### Official Review · AnonReviewer3 · 2018-11-11
**A good application paper addressing the (re)ranking problem with pointer networks**

**Rating:** 6
**Confidence:** 4

**Review:**

This paper formulates the re-ranking problem as a sequence generation task and tackles it with the pointer-network architecture. The paper formulates the problem clearly. The proposed model is trained on click-through log and outputs a ranking list of candidate items. The authors discuss two different potential approaches to train the model and conduct synthetic experiments as well as a real-world live experiment. The model also ran in a live experiment (A/B testing).

Some issues I concern about:

In Equation 7,  why do the authors introduce the baseline function into supervised learning?  I guess this is due to the advantage function in REINFORCE. But the authors should state the reason and correctness of this derivation.

In Section 4, implementation details part:
“R=NDGC@10” should be “R=NDCG@10”, (and maybe the authors could give a citation for each metric).
“baseline b(x) in Eq(3) and Eq(6)” should be “baseline b(x) in Eq(3) and Eq(7)”, and why do the authors use the same baseline function for REINFORCE and supervised learning?
Can the author specify why Seq2Slate with REINFORCE is not regularized while all other models are trained with L2- regularization?

In Section 4.1,  I think the authors can give a direct comparison between their models and [Ai, et al., SIGIR18]. Compared with [Ai, et al., SIGIR18], where the authors use more metrics (i.e. Expected Reciprocal Rank), there are fewer metrics used in this paper. Especially, I want to see the performance of Seq2Slate with REINFORCE (reward as NDCG) on other metrics.


In Section 4.2:
Why the authors do not conduct real-world data experiments on Seq2Slate with REINFORCE? I am wondering whether the time complexity of REINFORCE is too high for experiments on large scale datasets.
[Important] The authors stated that their model is sensitive to input order. However, it seems that they do not specify the input order of training sequences in Section 4.1. Is the order fixed? And in my opinion, the robustness of the model can be improved by shuffling the input sequence during the training stage, just like data augmentation in computer vision. I suggest the authors conduct the extended experiments.

General comments:

Quality (6/10): The paper uses the Seq2Seq architecture for the learning-to-ranking task, which is self-contained and has no obvious flaws. Experiments could be more perfect, especially, the author can add more metrics. The authors also give a comparison of their proposed training variants along with a detailed analysis of how their models can adapt to different types of data. In addition, a live experiment (A/B testing) is conducted.


Clarity (6/10): The paper is well written in general, some typos shall be fixed.

Significance (6/10): The authors validated their model on both synthetic data (based on two learning to rank benchmark datasets) and real-world data. I think the authors can use more metrics besides MAP and NDCG.

Originality (5/10): The paper can be regarded as an adaption of the Seq2Seq framework with pointer networks on learning-to-rank tasks. Although the authors give an analysis on why the traditional training approach (teacher forcing) cannot be applied to their tasks and give two alternative approaches, this paper stills seems to a direct application with minor innovation on training approaches and loss functions.

In summary, I think this is a good paper addressing the (re)ranking problem with pointer networks, but it is more suitable for conferences focusing on application and industry like SIGIR or KDD instead of the deep learning conference ICLR.

---

> ### Author Response · Authors · 2018-11-14
> **Clarifications**
>
> Thanks for the valuable feedback. We address specific points here and more general points in the common response.
>
> ## why do the authors introduce the baseline function into supervised learning? why do the authors use the same baseline function for REINFORCE and supervised learning?
> ===
> Policy gradients are computed from sampled sequences and are known to be noisy. For this reason we use a baseline function for both RL and supervised training in order to reduce variance. Although we used the same notation b(x) in both cases, the functions are in fact different: b(x) approximates the reward R for RL and the loss L in supervised learning. Thank you for pointing this out, we will use a different notation to make this distinction clear.
>
> ## Can the author specify why Seq2Slate with REINFORCE is not regularized
> ===
> We observed that reinforce training was not prone to overfitting, which is generally expected since the policy gradients are noisy.
>
> ## Why the authors do not conduct real-world data experiments on Seq2Slate with REINFORCE?
> ===
> We did not conduct experiments with REINFORCE on real-world data since it had worse performance compared to supervised training on the benchmark data and is generally more sensitive (e.g., high variance). In terms of runtime, RL needed 4x more time till convergence when training on the benchmarks, but training time was not a bottleneck in the real-world experiments.
>
> ## the author can add more metrics (i.e. Expected Reciprocal Rank)
> ===
> We provide here results for ERR@10 to add to Table 1:
>                                 Yahoo  Web30k
> Seq2slate                  0.86   0.77
> AdaRank                   0.83   0.61
> Coordinate Ascent  0.74   0.52
> LambdaMART          0.77   0.68
> ListNet                      0.70   0.73
> MART                        0.77   0.65
> Random Forests     0.74   0.60
> RankBoost               0.71   0.43
> RankNet                   0.80   0.73
> These results are consistent with the trends observed for the other metrics. We will include them in the revision.
>
> ## The authors stated that their model is sensitive to input order. However, it seems that they do not specify the input order of training sequences in Section 4.1. Is the order fixed? And in my opinion, the robustness of the model can be improved by shuffling the input sequence during the training stage, just like data augmentation in computer vision. I suggest the authors conduct the extended experiments.
> ===
> In Section 4.1 we specified that: “We select this base ranker by training all models in the RankLib package, and selecting the one with the best performance on each data set (MART for Yahoo and LambdaMART for Web30k).” We used those base rankers to order the inputs to seq2slate. In the “Sensitivity to input order” paragraph we describe experiments with “shuffled data”, where we feed inputs in random order. The results in Table 2 verify that ranking from scratch is harder, which is consistent with previous studies. We share your belief that seq2slate can learn to rank well even without relying on a base ranker, but this will require more training examples and will take longer to train. In this work we wanted to focus modeling capacity of seq2slate on interactions between items rather than individual items, so we decided to use the base ranker to order inputs. We agree that experiments with multiple reshuffling per example are interesting and will work to add them in the revision.

---

### Author Response · Authors · 2018-11-14
**Thanks for the feedback!**

We thank the reviewers for their valuable feedback. We address common points here and more specific points in individual responses.

For novelty, we would like to emphasize that sequential decoding and learning from click-through logs are central to our work and novel in this setting. Specifically, sequential decoding lets the score of items change depending on previously chosen items, thereby allowing the model to account for high-order interactions in a natural and scalable manner (see eq 1 and 2). This is one of the primary technical contributions of our paper, and we believe a novel, important contribution to sequence modeling for ranking. Indeed this approach can lead to much better results -- see Table 3. Further, we believe that training from click-through logs makes our framework much more interesting and useful than approaches that rely on ratings which are expensive to obtain (e.g., Ai et al. 2018). In addition, our results on gradient updates for sampled sequences (eq 6-7) are of general interest for training sequence models when the reward depends on the model parameters. Although the derivation itself is not complicated, these results provide new insights and, we believe, are non-trivial as evidenced by the fact that they were overlooked in all previous work of which we are aware (e.g., Bengio et al. 2015).

Regarding the work of Ai et al. (2018), we note that our work was done independently of this recently published work. Their approach is similar to our one-step decoding variant. The main differences between that approach and ours: training from relevance scores while we focus on click-through data, using GRU cells instead of LSTM cells in seq2slate, and reversing the input order. We see GRU and input order as orthogonal choices which could be also used within seq2slate and that the results in Tables 2 and 3 faithfully capture the differences between the approaches. That said, the discussion of the Ai et al. model and the contrast with ours in the current version of the paper could be more detailed. We’ll address that in the revision.

We point out that the seq2slate model is not a re-ranker per se, but applies equally to unordered candidate sets. We do note (as does other work) that seq2seq models are often affected by input order, so ordering (e.g., by a base ranker) can improve performance -- a fact confirmed by our experiments. It might be fair to say that it is especially well-suited to the re-ranking task, but not to insist that it is a re-ranker. We will rephrase to stress this point.

Regarding fit to ICLR, one way to see our work is for learning a representation of the list of candidate items for the purpose of ranking with a deep network (RNN) while modeling high-order interactions. This is very relevant to ICLR as evident by the large number of submissions on this topic, for example: “SET TRANSFORMER”, “BEYOND GREEDY RANKING: SLATE OPTIMIZATION VIA LIST-CVAE”, “LEARNING REPRESENTATIONS OF SETS THROUGH OPTIMISED PERMUTATIONS”, to name a few.

---

### Meta-Review · Area_Chair1 · 2018-12-17
**valuable application area, limited novelty**

**Confidence:** 4
**Recommendation:** Reject

**Metareview:**

The paper addresses the problem of learning to (re)rank slates of search results while optimizing some performance metric across the entire list of results (the slate). The work builds on a wealth of prior work on slate optimization from the information retrieval community, and proposes a novel approach to this problem, an extension of pointer networks, previously used in sequence learning tasks.

The paper is motivated by an important real world application, and has potential for significant practical impact. Reviewers noted in particular the valuable evaluation in an A/B test against a strong production system - showing that the work has practical impact. Reviewers positively noted the discussion of practical issues related to applying the work at scale. The paper was found to be clearly written, and demonstrating a thorough understanding of related work.

The authors and AC also note several potential weaknesses. Several of these were addressed by the authors, as follows. R3 asked for more breadth on metrics, and additional clarifications - the authors provided the requested information. Several questions were raised regarding the diverse-clicks setting and choice of hyperparameter \eta - both were discussed in the rebuttal. Further analysis / discussion of computational and performance trade-offs are requested and discussed.

Overall, the main drawback of the paper, raised by all three reviewers, is the size of the contribution. The paper extends an approach called "pointer networks" to the model application setting considered here. The reviewers and AC agree that, while practically relevant and interesting, the research contribution of the resulting approach limited. As a result, the recommendation is to not accept the paper for publication at ICLR in its current form.